# Optimization of High Hydrostatic Pressure Treatments on Soybean Protein Isolate to Improve Its Functionality and Evaluation of Its Application in Yogurt

**DOI:** 10.3390/foods10030667

**Published:** 2021-03-20

**Authors:** Chenxiao Wang, Hao Yin, Yanyun Zhao, Yan Zheng, Xuebing Xu, Jin Yue

**Affiliations:** 1Bor S. Luh Food Safety Research Center, SJTU-OSU Innovation Center for Environmental Sustainability, Key Laboratory of Urban Agriculture, Ministry of Agriculture, Shanghai Jiao Tong University, Shanghai 200240, China; wangcx01@sjtu.edu.cn (C.W.); yinhao12138@sjtu.edu.cn (H.Y.); yanyun.zhao@oregonstate.edu (Y.Z.); 2Department of Food Science and Technology, Oregon State University, 100 Wiegand Hall, Corvallis, OR 97331, USA; 3Wilmar Global Research and Development Centre, No. 118 Gaodong Rd., Shanghai 200137, China; zhengyan1@cn.wilmar-intl.com (Y.Z.); xuxuebing@cn.wilmar-intl.com (X.X.)

**Keywords:** high hydrostatic pressure, soybean protein isolate, functional properties, soy yogurt

## Abstract

This work aimed to improve the functional properties of soybean protein isolate (SPI) by high hydrostatic pressure (HHP) and develop SPI incorporated yogurt. Response surface methodology (RSM) was used to optimize the HHP treatment parameters, including pressure, holding time, and the ratio of SPI/water. Water holding capacity, emulsifying activity index, solubility, and hardness of SPI gels were evaluated as response variables. The optimized HPP treatment conditions were 281 MPa of pressure, 18.92 min of holding time, and 1:8.33 of SPI/water ratio. Water and oil holding capacity, emulsifying activity, and stability of SPI at different pH were improved. Additionally, relative lipoxygenase (LOX) activity of HHP treated SPI (HHP-SPI) was decreased 67.55 ± 5.73%, but sulphydryl group content of HHP-SPI was increased 12.77%, respectively. When incorporating 8% of SPI and HHP-SPI into yogurt, the water holding capacity and rheological properties of yogurt were improved in comparison with yogurt made of milk powders. Moreover, HHP-SPI incorporated yogurt appeared better color and flavor.

## 1. Introduction

Soy is a source of the predominant vegetable proteins. Soy protein isolates (SPI) are generally produced from soybean by being collected as precipitated curd in the acidic condition [1], and SPI are essential for a wide range of protein-based food formulations [2,3], owing to their outstanding processing ability, high nutritional value, and low cost. The functional properties of SPI, such as solubility and emulsifying properties, are determined by the protein composition, structure, degree of denaturation, and aggregation. According to the needs in the specific food system, those properties can be modified by chemical, physical, or enzymatic methods [4].

High hydrostatic pressure (HHP) technology is a non-thermal food-processing method showing potential for the development of new food products with additional functional and health benefit [5]. Under HHP, proteins are regulated by the Le Chatelier’s principle and shifted to a lower volume conformer, which in turn changes their structure and conformation and influences their functional properties [6,7]. A number of researchers have studied the impact of HHP treatment on SPI in recent years and found that solubility, water holding capacity, and foaming property could be improved under medium-high pressure (up to 400 MPa) for short processing time (5–20 min), while these properties tend to decline under ultra-high pressure (above 400 MPa) [8,9,10]. Li et al. confirmed that the HHP modified SPI could have potential utilization in infant formula, lying in the better swallowing properties, in vitro digestibility, and lower allergenicity [11]. Other researchers found that protein solubility and in colloidal solubility of SPI are improved, and colloidal-stable calcium added SPI dispersions can be obtained by HHP treatment, since pressure promotes the formation of calcium-protein species that could establish bridges between the droplets [12].

Plant protein replaced food products have received great attention lately due to the increasing consumer awareness about the impacts of animal-based food production on the environment and health benefits of plant protein over animal protein, e.g., lowering blood cholesterol level [12,13]. Soybean has been especially a popular source for plant-based yogurt recently, because of its accessibility and quality. Previous studies suggested that soy-based media might possess sufficient substrates to promote the growth of probiotic microorganisms [14,15]. However, the main problems with this type of product are appearance and texture caused by phase separation and the off-flavor of the plant material [16]. Besides combinations of gelling agents, improving the functional properties of proteins, especially increasing the stabilization of soy proteins on acidified food systems, may be a solution [17]. For the flavor of soybean products, controlling its Lipoxygenase (LOX) activity is very important, since it is responsible for the generation of a series of off-flavor components in soybean and soy-derived products. As mentioned earlier, modified by HHP can improve the functional properties of SPI, thus improving the quality of soybean products. To the best of our knowledge, this research revealed an application of modified soy proteins in yogurt for the first time.

The main purpose of this work is to use HHP treatment to improve the functional properties of SPI and apply it in producing soy-based yogurt. Response surface methodology was used to optimize the HHP process to modify the functional properties of SPI, including solubility, water holding capacity (WHC), and emulsifying activity index (EAI). A SPI-milk mixed yogurt was then produced, and its physicochemical and rheological properties, as well as the volatile flavor compounds characteristics, were evaluated.

## 2. Materials and Methods

### 2.1. Materials

Soy protein isolate (SPI) was provided by Yihai Kerry Group (Shanghai, China) and protein content (dry basis) was 85.1%, respectively. Whole milk powder (protein 24.0%, fat 28.4%, and carbohydrate 39.0%) was purchased from Fonterra Co-operative Group (Auckland, New Zealand). 1-Anilino-8-naphthalenesulfonate (ANS) and 5, 5-dithiobis (2-bitrobenzoic acid) (DTNB) were purchased from Sigma-Aldrich (St. Louis, MO, USA).

### 2.2. High Hydrostatic Pressure (HHP) Treatments

HHP treatments were performed using a High Pressure Iso-Lab System (FPG7100, Stansted Fluid Power, Stansted, Essex, UK) and a hydraulic type cell with an inner capacity of 1 L and a water jacket for temperature control. A mixture of propylene glycol and water (30:70) was used as pressure-transmitting medium. SPI solution (SPI/water ratio was 1:4, 1:8 or 1:12) was prepared by dissolving the powder in distilled water and stirring for 2 h. Then, 360 g SPI solution is vacuum-packed and subjected to HHP treatments at pressure of 200 ℃ 400 MPa, holding time of 10–30 min. The processing temperature was set at 25 ℃ [18]. After HHP treatment, the SPI solutions were freeze-dried (Triad, Labconco, Kansas city, Missouri, USA) and stored at −20 ℃ for further analysis.

### 2.3. The Response Surface Methodology (RSM) Design

RSM is useful for optimizing, designing, developing, and improving processes where the responses are affected by several variables. Moreover, interactions between variables can be identified and quantified by RSM and its widely used in food industry processes for optimizations. The effects of three independent variables (pressure (MPa, *X*_1_), holding time (min, *X*_2_) and material-liquid ratio (*X*_3_)) on three response variables (water holding capacity (g/g, Y_1_), emulsifying activity index (m^2^/g, Y_2_), and solubility (%, Y_3_) were evaluated using Box–Behnken design (BBD), Table 1. The software Design Expert 8.0 (Stat-Ease Inc., Minneapolis, MN, USA) was used to analyze the collected data. The effects of the independent variables *X*_1_, *X*_2_, and *X*_3_ on the response value *R*^2^ were evaluated using the empiric second-order polynomial regression model as Equation (1):(1)$$R=\beta_{0}+\overset{3}{\underset{i=1}{\sum }}\beta_{i}X_{i}+\overset{3}{\underset{i=1}{\sum }}\beta_{ii}X_{i}^{2}+\sum^{\;}\overset{3}{\underset{i<j=1}{\sum }}\beta_{ii}X_{i}X_{j}$$

### 2.4. Determination of the Functional Properties of SPI

#### 2.4.1. Solubility

The solubility of SPI was determined according to the method of Condés et al. with minor modifications [19]. In brief, 200 mg of SPI was added into 20 mL of distilled water (pH 7). The mixture was agitated at room temperature for one hour at 25 ℃, and then centrifuged at 10,000 *g* for 20 min. The mass of resolved proteins in the supernatant was determined by micro-Kjeldahl method (N = 6.25). The protein solubility was calculated as Equation (2):(2)$$\mathrm{Solubility}\;\left(\%\right)=P\times 100/P_{total}$$
where *P* was the mass of resolved protein in the supernatants and *P_total_* was the total protein content.

#### 2.4.2. Water Holding Capacity (WHC) 

The water holding capacity of SPI was determined using the method described by Ogunwolu et al. with minor modifications [20]: 1.5 g of protein sample was mixed with 15 mL of distilled water (pH 7) and centrifuged at 8000 *g* for 10 min at 25 ℃. The supernatant was then removed and the residue was weighted. WHC was calculated as Equation (3):(3)$$\mathrm{WHC}\;\left(g/g\right)=(W_{2}-W_{1})/W_{0}$$
where *W*_2_ was the weight of centrifuge tube and precipitated protein after absorbing water; *W*_1_ was the weight of centrifuge tube and protein sample; *W*_0_ was the weight of protein.

#### 2.4.3. Emulsifying Properties

The EAI and emulsifying stability index (ESI) of SPI were determined according to the method of Pearce and Kinsella with some modifications, respectively [21]: 2 g/L SPI solutions were prepared using acetate buffer (0.05 M, pH 3), distilled water, or PBS (0.05 M, pH 8). Then, 3 mL of soybean oil and 9 mL of the SPI solutions (pH 3, 7, or 8) were mixed and homogenized in a high-speed homogenizer (Model 420, Thermo Fisher Scientific (China) Co., Ltd.) for 1 min at 4000 rpm to form the emulsion. A 50 μL of emulsion was taken from the bottom of the emulsion immediately (0 min) or at 10 min after homogenization and diluted in 5 mL of sodium dodecyl sulfate solution (0.1%, *w*/*v*). The absorbance of the produced emulsions was measured at 500 nm using a spectrophotometer (UV-1800, Shimadzu International Trading (Shanghai) Co., Ltd.). EAI and ESI were calculated by the following Equations (4) and (5), respectively:(4)$$\mathrm{EAI}\;\left(m^{2}/g\right)=\frac{2\times 2.303}{c\times \left(1-\varphi \right)\times 10000}\times A_{0}\times DF$$
(5)$$\mathrm{ESI}\;\left(\min\right)=A_{0}/A_{10}\times 10\;$$
where *c* was the initial concentration of protein, φ was the fraction of oil used to form the emulsion, *DF* was the dilution factor, and *A*_0_ and *A*_10_ were the absorbance of the diluted emulsions at 0 and 10 min, respectively.

### 2.5. Determination of the Physicochemical Properties

#### 2.5.1. Surface Hydrophobicity (H_0_) 

The surface hydrophobicity was analyzed according to the method of Yang et al. using 1-anilino-8-naphthalene-sulfonate (ANS) as a fluorescence probe [22]. SPI dispersion (1 mg/mL) were centrifuged at 10,000 rpm and 4 ℃ for 20 min. The protein concentration in the supernatants was measured according to the Lowry method [23]. The supernatant was serially diluted with deionized water to obtain protein concentrations ranging from 0.005 to 0.5 mg/mL. A 50 μL of ANS (8.0 mM) was then added to 4 mL of protein solutions, respectively. The relative fluorescence intensities of the ANS-proteins conjugates were measured at room temperature using a fluorescence spectrophotometer (Hitachi F4500, Tokyo, Japan) at wavelengths of 365 nm (excitation) and 484 nm (emission), with a constant excitation and emission slit of 5 nm. The protein hydrophobicity was expressed as the initial slope of relative fluorescence intensity versus protein concentration (mg/mL) (calculated by linear regression analysis).

#### 2.5.2. Sulphydryl Group Content

Sulfhydryl group content of soybean proteins was measured according to the method of Beveridge et al. using Ellman’s reagent [24]. Ellman’s reagent was prepared by dissolving 40 mg of DTNB in 10 mL of PBS (0.1 M pH 8.0). One milliliter of SPI dispersions (1 mg/mL) was added into 5 mL of Tris-Gly buffer (0.086 mol Tris and 0.09 mol Gly, pH 8.0). Subsequently, 40 μL of Ellman’s reagent was added for color reaction. The mixture was shaken and incubated at room temperature for 10 min, and the absorbance was then measured at 412 nm with a UV spectrophotometer. Sulphydryl group content was calculated as Equation (6):(6)$$\mathrm{SH}\;\mathrm{content}\;\left(\mu \mathrm{mol}/g\right)=\frac{10^{6}\times \mathrm{DF}}{c\times 13600}\times A_{412}$$
where *c* was the initial concentration of protein, DF was the dilution factor, and *A*_412_ was the absorbance of the diluted emulsions, respectively.

#### 2.5.3. Lipoxygenase (LOX) Activity

LOX activity of SPI was measured according to the method of Li et al., with minor modifications [25]. Briefly, an 8.0 mg/mL of SPI dispersions was prepared and centrifuged at 10,000× *g* and 4 ℃ for 20 min. The supernatant was then reacted with an enzyme substrate prepared previously. Right after mixing, the absorbance was measured at 234 nm and 25 ℃ by a UV spectrophotometer. The residual activity (RA) of LOX was calculated as Equation (7):(7)$$\mathrm{RA}=\;\frac{A}{A_{0}}\times \;100$$
where *A* was the LOX activity of SPI after HHP treatment, and *A*_0_ was the initial LOX activity before treatment.

#### 2.5.4. Circular Dichroism (CD) Spectra Analysis

A protein concentration of 0.1 mg/mL was selected for CD spectra analysis [22]. The sample was scanned at the far UV range (240–190 nm) at room temperature with a JASCO J-815 spectropolarimeter at the scan speed of 50 nm/min, band width of 1.0 nm, and response time of 0.25 s. The CD spectra were expressed as mean residue ellipticity (deg·cm^2^/d mol).

### 2.6. Preparation of SPI Incorporated Yogurt

The yogurt was prepared by reconstituting milk powder (15% [*w*/*v*]) and untreated or optimized HHP treated SPI (8% [*w*/*v*]) in distilled water under continuous stirring for 30 min and then homogenized in high-speed homogenizer at 10,000 rpm. The mixtures were heated at 95 ℃ for 10 min, then cooled down to 42 ℃, and finally inoculated with 0.2% (*w*/*v*) deep frozen commercial cultures (Baishengyou, Thankcome, Suzhou, China) of Lactobacillus bulgaricus and Streptococcus thermophilus in water bath (42 ℃) for 7 h. The yogurt was then stored at 4 ℃ for 24 h for quality evaluation. The yogurt made from milk powder (15% [*w*/*v*]) using the same procedure was used as a control.

### 2.7. Determination of Physicochemical Properties of the SPI Incorporated Yogurt

After storage, the pH of yogurt samples was determined using a pH meter (Seven-Multi; METTLER TOLEDO Instruments, Shanghai, China). The titration acidity (TA) was measured according to the titration method of Silva et al. [26]. WHC was determined using same method as described in Section 2.4.2 with a different centrifuge speed of 480 *g* for 10 min at 20 ℃.

The color of samples was determined using a colorimeter (Labscan XE; Hunter laboratory, Reston, VA, USA), in which CIEL * a * b * system. L *, a *, and b * were recorded, and ΔE (total color difference) was calibrated by taking the control sample as the reference [27].

The rheological properties of SPI-milk yogurts were evaluated at 25 ± 0.1 ℃ according to the method described by Mei et al., using a Haake RheoStress 6000 rheometer (Thermo Scientific, USA) [28]. A cylinder and plate geometry (12.50 mm diameter, 50.00 mm length, and 2.00 mm gap) was employed in all measurements. Flow curves were performed upward and downward with the shear rates ranging from 0.1 to 100 s^−1^. The Ostwald deWaele model was used to fit the experiment data, and was represented by the Equation (8):(8)$$\eta_{\alpha }=\frac{\tau }{\gamma }=k\gamma^{n-1}$$
where *η_a_* was the apparent viscosity (Pa·s), τ was the shear stress (Pa), γ was the shear rate (s^−1^), k was the consistency index (Pa·s^n^), and *n* was the flow behavior index.

### 2.8. Flavor Analysis of the SPI Incorporated Yogurt

Flavor is one of the most important properties of yogurt products, and volatile components leading to off-flavors can cause the product to be unsatisfactory for the tastes of consumers [29]. The yogurt sample was added into a glass vial (10 mL). Volatile compounds were extracted using Head Space Solid Phase Micro Extraction (HS-SPME) technique, and separated using a DB-Wax column (30 m × 250 μm × 0.25 μm; J and W Scientific, Folsom, CA, USA). Desorption of the extracted volatiles was carried out using a GC-MS system (Agilent 7890B-5977B, Agilent Technologies, Santa Clara, CA, USA) by the splitless mode. Helium was used as carrier gas at a flow rate of 1.0 mL/min, and the MS was operated in scan mode. The oven temperature was held at 40 ℃ for 4 min (desorption period), increased to 250 ℃ with a rate of 5 ℃·min^−1^, and then held at 250 ℃ for 5 min. The total run time was 50 min. The NIST 2011 mass spectral library (Gaithersburg, MD, USA) was used to identify the volatile compounds.

### 2.9. Statistical Analysis

Data were analyzed using SPSS (version 23.0 for Mac, SPSS Inc., Chicago, IL, USA) following an analysis of variance (ANOVA) one-way linear model and reported as mean values and standard deviations. Mean comparisons were performed using the Duncan test, and the significance level was established for *p* < 0.05.

## 3. Results and Discussion

### 3.1. Optimization of HHP Parameters by RSM

Table 1 reports the results obtained from 17 experimental runs following the BBD design. *R*^2^ values of all models were higher than 0.9, indicating that more than 90% of the changes could be expressed by these models. The coefficients of the models are shown in Appendix A calculated using ANOVA analysis. The experimental data were processed using quadratic regression polynomial analysis and fitting, and the fitted quadratic models for WHC (Y_1_), EAI (Y_2_), and solubility (Y_3_) in coded variables are given in (Equations (9)–(11)), respectively. The *p*-values of all models were less than 0.01 (Appendix A). Furthermore, the *p*-values of lack of fit were higher than 0.05 in all models, confirming the validity of models. These results suggested that three models had good fitting accuracy and could be used for the optimization design. Moreover, three-dimensional response surface plots and curves (Figure 1) were established to further illustrate the predicted optimal inclusion conditions for HHP treatments.

(9)$$Y_{1}=5.10-0.34X_{1}-0.16X_{2}+0.17X_{3}+0.22X_{1}X_{3}+0.30X_{2}X_{3}-0.25X_{1}^{2}-0.40X_{2}^{2}-0.38X_{3}^{2}$$

(10)$$Y_{2}=26.68-3.21X_{1}-0.16X_{2}-0.52X_{3}-3.74X_{1}^{2}-2.38X_{2}^{2}-3.81X_{3}^{2}$$

(11)$$Y_{3}=31.64-0.30X_{1}+1.50X_{2}+4.18X_{3}-3.32X_{2}X_{3}-9.03X_{1}^{2}-3.98X_{2}^{2}-5.53X_{3}^{2}$$

Figure 1A demonstrated that water holding capacity increased with the pressure elevated from 200 to 300 MPa, and then decreased. The initial pressure increasing could cause the partial unfolding of the protein, which allowed interactions between the subunits to form a flexible network where water was entrapped, thus increasing WHC [8]. When the pressure or the holding time continued increasing, proteins aggregation occurred and declined the WHC (Figure 1A). These results were coincident with the study by Li et al. [11]. They treated 1% of SPI under HP and found that the highest WHC was achieved when pressure was below 300 MPa for 15 min, whereas higher pressure or longer holding time lead to decreasing it. The same trends of changing WHC were observed in other studies. For example, Molina et al. found that WHC of 20% (*w*/*v*) of SPI reached the highest point when treated at 500 MPa, but decreased when pressure further increased [30].

The effects of pressure (X_1_), time (X_2_), and SPI/water ratio (X_3_) on emulsifying activity index (EAI) were illustrated through response surfaces and contour plots in Figure 1B. EAI initially increased with higher level of pressure (X_1_) and time (X_2_) due to the partial or total denaturation of SPI enhancing the surface activity. Low ratio of SPI/water (X_3_) during the HHP treatment also reduced EAI, probably owing to the interaction between the subunits of protein. Molina et al. suggested that 10% (*w*/*v*) 11S globulins of SPI showed the optimum value of EAI after treatment at 200 MPa for 15 min, while 7S globulins and SPI at the same concentration showed the highest EAI after treating at 400 MPa for 15 min [31].

Figure 1C exhibited the effects of pressure (X_1_), time (X_2_), and SPI/water ratio (X_3_) on SPI solubility. At the beginning, when the pressure and holding time increased, the solubility increased to around 30% due to the change of SPI tertiary structure enhancing the protein–solvent interactions. As the pressure and holding time continued to rise, the solubility went down in all the trials. Figure 1C suggests that low ratio of SPI/water significantly decreased the SPI solubility, since the unfolding of the globulins and the exposing of hydrophobic groups and SH groups might lead to aggregation of SPI to form the insoluble residues [32]. Wang et al. reported that HPP processing of SPI at 200 MPa and pH 6.8 for 15 min enhanced its solubility, whereas the solubility declined at 400 MPa and then rose again after 600 MPa treatment [9].

The optimization of the HHP treatment was chosen based on the prerequisite of fulfilling the weight of four response values as 1:1:1:1 by applying the optimum formulation point of numerical method generated by Design Expert software. The optimum conditions for the HHP treatments were calculated to be at pressure (X_1_) = 281.09 MPa, time (X_2_) = 18.92 min, and SPI/water ratio (X_3_) = 1:8.33. Under these conditions, the maximum WHC, EAI, and solubility were 5.19 g/g, 27.06 m^2^/g, and 31.6%, respectively. In order to validate the reasonability of the model equations, a treatment of SPI was carried out in triplicate under the optimal conditions. The WHC, EAI, solubility, and hardness of gel were 5.35 ± 0.04 g/g, 28.11 ± 1.67 m^2^/g, and 33.7 ± 0.22%, respectively, which was in good agreement with the predicted data.

### 3.2. Effect of HHP on Properties of SPI

#### 3.2.1. Effect of HHP on Physicochemical Properties of SPI

Several physicochemical properties, including surface hydrophobicity, sulphydryl group content, lipoxygenase (LOX) residual activity, and circular dichroism (CD) spectra of the untreated and HPP treated (281 MPa) SPI were shown in Figure 2 The surface hydrophobicity rose from 683.47 to 812.64 after HHP treatment (Figure 2A). Li et al. found similar trends of surface hydrophobicity of SPI treated by low pressure (e.g., 200 MPa) [33], and suggested that the increase was caused by the unfolding of the protein with the exposure of some hydrophobic groups into the medium under pressure. Xi and He pointed out that an upturn of free SH groups usually indicates protein unfolding and back-bone fragmentation, whereas a reduction indicated a cross-linking effect on protein. Pressure caused a slight increase of the content of sulphydryl groups (Figure 2D), suggesting that high pressure might trigger the interchange reaction between sulphydryl groups and disulfide bonds, thus forming small particles and fragments of protein. LOX is an enzyme responsible for the generation of a series of off-flavor components in soybean and soy-derived products [34], and its residual activity was reduced to 67.55 ± 5.73% after 281 MPa treatment (Figure 2C).

Circular dichroism (CD) spectra of untreated and HHP treated SPI were shown in Figure 2D. As described by Puppo et al. [8], SPI presented a typical spectrum of α+β proteins, which showed a positive band below 195 nm with a zero crossing around 195 nm, and a negative band near 210 nm. After HHP treatment, a significant rise at around 225 nm can be observed, which could be due to the change of secondary structure of SPI from α-helix to β-sheet and random coil. We have calculated that after 200 MPa treatment, the α-helix content decreased to 23%, whereas the content of β-sheets and random coils increased to 16% and 7%, respectively. These results were consistent with the reports by Li et al., using β-lactoglobulin, which found an increment of random coil content and a reduction of α-helix content after 400 or 600 MPa for 30 min treatment [35].

#### 3.2.2. Effect of HHP on Functional Properties of SPI

The functional properties of the untreated and HHP treatment (281 MPa) SPI, including water holding capacity, solubility, and emulsifying properties in different pH were shown in Figure 3. After HHP treatment, the WHC and solubility of SPI were significantly improved. The partial unfolding of the protein under high pressure not only allowed interactions between the subunits to form a flexible network in which water was entrapped, but enhanced the protein–solvent interactions, thereby enhancing the WHC and solubility [36]. Manassero et al. reported that combined thermal-high hydrostatic pressure treatment improved the solubility and physical stability of soy protein [18].

The change of emulsifying activity index (EAI) and emulsifying stability index (ESI) of SPI produced by HHP treatment showed significant difference with different pH conditions (Figure 3C,D). EAI and ESI described the capability of proteins to contribute to the formation of and stabilization of dispersion systems. Surface hydrophobicity, solubility, and the resulting capability to decrease the interfacial tension were proven to be crucial for the formation of the emulsion, while molecular flexibility and interaction on the interface were decisive for emulsion stability [32]. EAI was higher at pH 8 because of the increase of solubility, as the solubilized protein could rapidly adsorb at the oil/water interface, facilitate the formation of densely packed films around the oil droplet, and help forming emulsion. Wang et al. concluded that HHP treatment remarkably increased the EAI values of SPI at 200 MPa, while a further increase in pressure (400 and 600 MPa) did not result in a significant change in EAI [9]. HHP treatment increased EAI at pH 3 because of the improvement of solubility, indicating a better emulsifying capacity of HHP–SPI in low pH system. Puppo et al. reported high-pressure processing (200 or 400 MPa for 10 min) seemed to improve emulsifying properties of SPI that have declined due to acidification at pH 3 [37]. Meanwhile, ESI increased at various pHs after HHP treatment, as the increasing of ζ-potential by HHP treatment may indicate an effective electrostatic repulsion between the droplets.

### 3.3. Yogurt Characteristics

Physicochemical properties, including final pH, titration acidity (TTA), WHC, and color, of three kinds of yogurt samples were reported in Table 2. Accordingly, the addition of SPI increased pH and TA, which was consistent with the result of Mohammadi et al. [38], but there was no significant difference between products made with SPI and HHP–SPI. The water in the yogurt after long-term storage might lose due to the intrinsic instability of gels or passive diffusion, which is described as syneresis. WHC is a critical parameter in evaluation yogurt, since it indicates the ability to keep serum in the gel structure [39]. The addition of SPI significantly increased the WHC of yogurt from 56.37 ± 1.71 to 83.42 ± 0.96 (SPI) and 90.22 ± 2.83 (HHP-SPI). Higher WHC of the HHP-SPI yogurt indicated the improved stability of SPI under acidic conditions and a more branched yogurt microstructure [28]. This result was consistent with the rise of WHC of SPI after HHP treatment as mentioned in Figure 3A.

A darker and yellower color of the yogurt was clearly observed by the naked eye in SPI and HHP-SPI yogurt samples. Correspondingly, the L * (lightness) values were lower in those two samples compared with control. However, HHP-SPI yogurt had higher L * value than SPI yogurt, which might be explained by the particle dispersion after HHP that result in light reflection on the yogurt gel surface [40]. The b * (yellow–blue axis) values were positive in all samples and samples with SPI had highest b* values. The ΔE* values of the SPI addition samples were >3, indicating that the color variation could be identified by the naked eye.

Figure 4 showed flow curves of yogurts with shear stress versus shear rate and Table 3 showed the results of the Ostwald deWaele model fitting. All samples showed hysteresis loops and shear thinning (thixotropic) behavior, which indicated the required energy for splitting gel structure in soy yogurt. Similar observations for soy yogurt were reported previously [41,42]. HHP-SPI yogurt showed the highest shear stress, which might suggest a stronger gel structure with higher resistance to shear forces, since HHP-SPI showed higher H_0_ and SH content that formed stronger gels. The curves were fitted to the Ostwald deWaele model and the values of the flow behavior index (n) and the consistency index (k) were obtained. The flow behavior index (n) was a measure of deviation of shear thinning fluids from Newtonian flow; meanwhile, the consistency index (k) was believed to be related to product acceptability for yogurt [43]. The experimental data fitted well to the model with *R*^2^ values generally above 0.98. Low n values (n < 1) were recorded in all samples, and the addition of SPI decreased the n value, indicating an increase of the pseudoplastic behavior of yogurt. It was regarded that higher pseudoplasticity could lead to better consumer acceptance of the product [44]. Furthermore, the addition of SPI and HPP-SPI both increased the k value of yogurt significantly, indicating that the products were more viscous.

The volatile compounds of three yogurt samples were shown in Figure 5. Typical volatile compounds of yogurt aroma and flavor included volatile acids, such as acetic, propionic, and butyric, and carbonyl compounds, such as acetaldehyde, acetone, acetoin, and diacetyl [29]. On the other hand, several researchers have identified pentanal, hexanal, heptenal, ethanol, octen-3-ol, and 2-pentylfuran as the main volatile compounds of soymilk flavor, and the main off-flavor compound produced by soymilk oxidation was C6 alkyl or alkyl aldehydes [45,46,47]. As shown in Figure 5A, the addition of SPI significantly reduced the formation of acetaldehyde, acid, and ketone compounds compared to control, indicating the contraction of yogurt aroma and flavor. However, more hexanoic acid (contributing to soy flavor also) was identified. Five volatile compounds relating to the beany flavor, namely 2-n-pentylfuran, hexanal, pentanal, 1-hexanol, and 1-octen-3-ol, were detected only in two SPI yogurt samples, and fewer of those compounds in HHP-SPI yogurt were detected (Figure 5B). LOX catalyzes the oxidation of PUFAs, leading to the production of aldehydes, ketones, and other volatile compounds that lead to off-flavor [48]. HHP treatment weakened the LOX activity of SPI; thus, the HHP-SPI yogurt contained fewer volatile compounds that provided off-flavor. Zhou et al. found that the yields of volatile compounds responsible for the beany flavor in LOX-lack soy yogurt were greatly decreased, and LOX-lack soy yogurt had the best sensory acceptability [34].

## 4. Conclusions

This study demonstrated that the functional properties of SPI could be significantly improved by HHP treatments. The identified optimal HHP conditions were 281 MPa pressure, 19 min holding time, and 1:8.3 of SPI/water ratio based on the response surface methodology. The HHP treated SPI exhibited higher surface hydrophobicity and sulphydryl groups content, and lower LOX activity. Water holding capacity and emulsifying activity index at pH 3 of SPI were significantly improved by HHP treatment. The yogurt with HHP-SPI displayed better water holding capacity and lighter color comparing with SPI incorporated one. In addition, the content of five volatile components relating to the beany flavor decreased in HHP-SPI yogurt, demonstrating the superior flavor. HPP is a promising technology to improve the functionality of SPI, which could be a superior plant protein for diary product.

## Figures and Tables

**Figure 1 foods-10-00667-f001:**
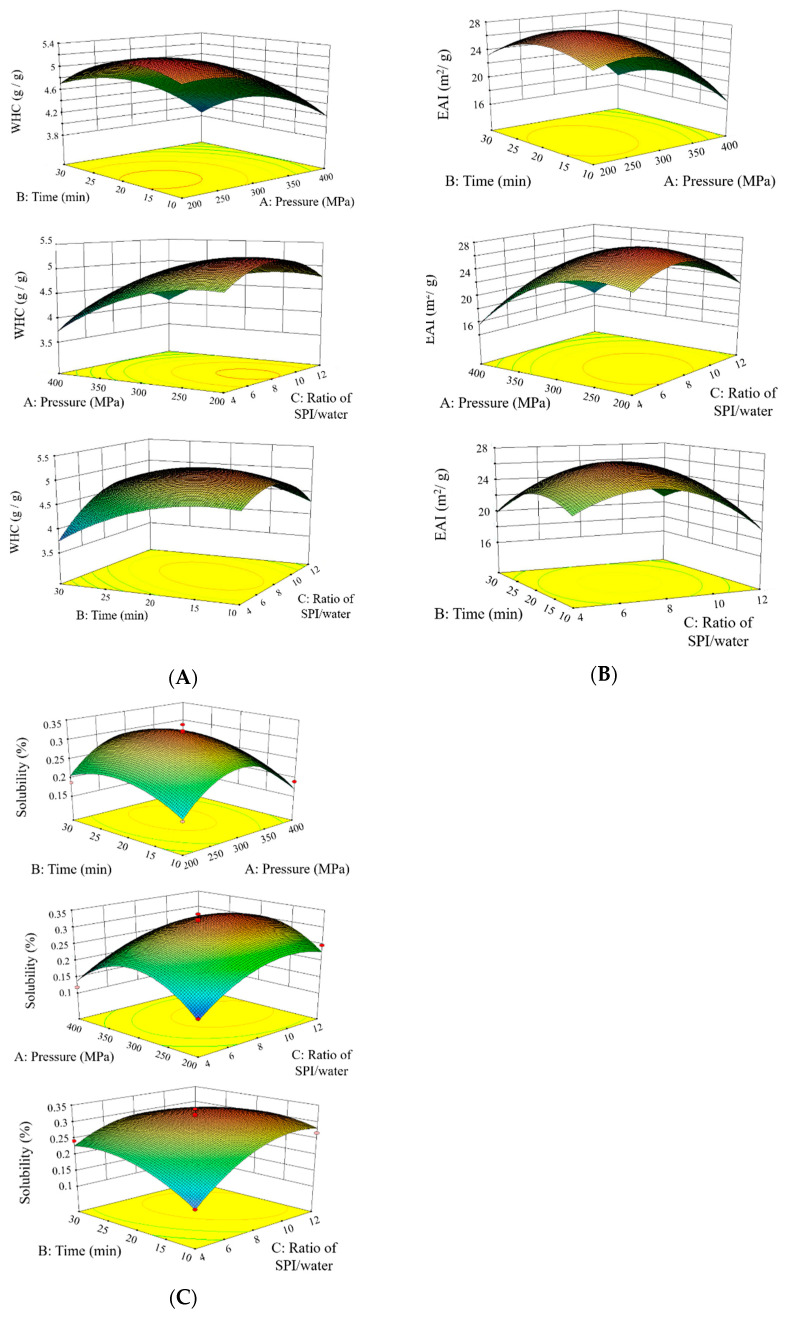
Response surface and contour plots for water holding capacity (WHC, **A**), emulsifying activity index (EAI, **B**), and solubility (**C**).

**Figure 2 foods-10-00667-f002:**
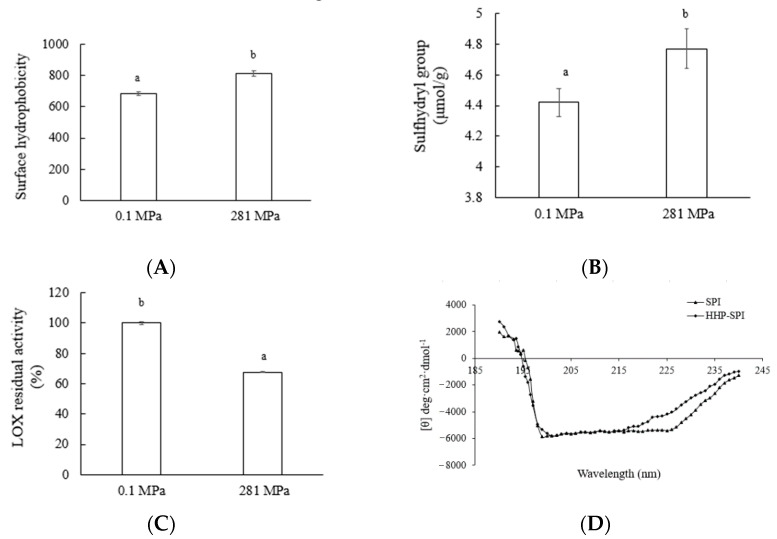
Physicochemical properties of soy protein isolate (SPI) (0.1 MPa) and high pressure treated SPI (HHP-SPI) (281 MPa) soybean proteins: surface hydrophobicity (**A**), sulphydryl group content (**B**), LOX residual activity (**C**), and CD spectra (**D**). Lowercase letters: statistical significance between groups.

**Figure 3 foods-10-00667-f003:**
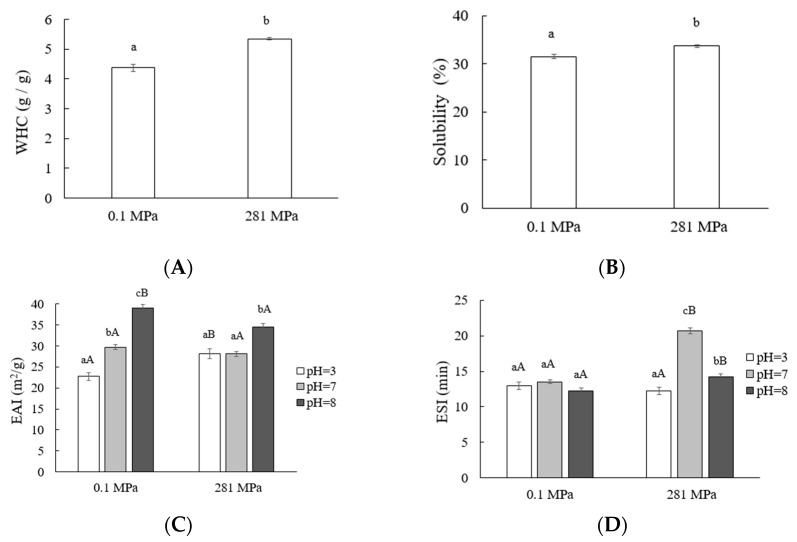
Functional properties of SPI (0.1 MPa) and HHP-SPI (281 MPa) soybean proteins: water holding capacity (**A**), solubility (**B**), emulsifying activity (**C**), and emulsifying stability (**D**). Capital letters: statistical significance between groups; lowercase letters: statistical significance in the groups.

**Figure 4 foods-10-00667-f004:**
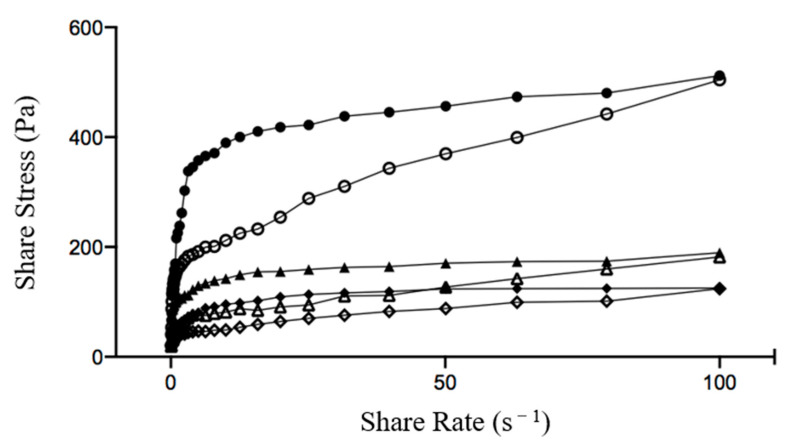
Flow curves and model fitting results of yogurt without SPI (control, ♦, ◊), or with SPI (▲, △) and HHP-SPI (●, ○). Shear rate was first increased (♦, ▲, ●) and then decreased (◊, △, ○).

**Figure 5 foods-10-00667-f005:**
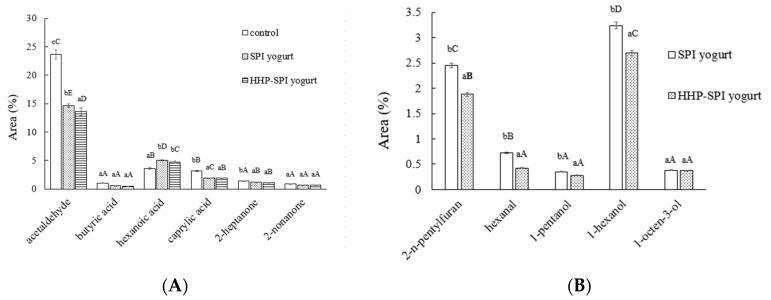
The volatile compounds of yogurt without SPI (control), or with SPI and HHP-SPI: (**A**) beany flavor components; (**B**) acetaldehyde, acid, and ketone components. Capital letters: statistical significance between groups; lowercase letters: statistical significance in the groups.

**Table 1 foods-10-00667-t001:** High hydrostatic pressure treatment conditions for Box–Behnken experimental design and obtained functional properties of soy protein isolate powder.

Order	Pressure (MPa)	Time (min)	SPI/Water Ratio	Water Holding Capacity (g/g)	Emulsifying Activity Index (m^2^/g)	Solubility (%)
1	200	20	1:4	4.63 ± 0.19	24.10 ± 0.52	12.6 ± 1.2
2	400	30	1:8	3.91 ± 0.05	18.40 ± 0.59	20.1 ± 1.7
3	400	20	1:4	3.77 ± 0.15	13.86 ± 1.24	11.8 ± 0.4
4	300	30	1:12	4.43 ± 0.29	20.33 ± 1.49	24.5 ± 0.8
5	300	20	1:8	5.04 ± 0.05	27.34 ± 0.93	33.8 ± 4.1
6	300	20	1:8	5.26 ± 0.31	27.13 ± 1.21	30.6 ± 2.3
7	200	30	1:8	4.88 ± 0.65	22.02 ± 1.24	18.7 ± 1.9
8	200	10	1:8	4.98 ± 0.15	23.49 ± 1.01	16.7 ± 3.3
9	300	10	1:12	4.37 ± 0.32	18.26 ± 0.31	26.7 ± 2.7
10	300	10	1:4	4.80 ± 0.47	22.64 ± 0.39	13.1 ± 1.6
11	300	20	1:8	5.02 ± 0.48	25.40 ± 0.12	29.6 ± 4.5
12	300	30	1:4	3.67 ± 0.11	20.75 ± 0.11	24.2 ± 2.6
13	300	20	1:8	5.11 ± 0.26	25.70 ± 0.87	31.9 ± 2.8
14	400	10	1:8	4.03 ± 0.03	18.37 ± 1.24	19.0 ± 1.2
15	300	20	1:8	5.09 ± 0.28	27.85 ± 1.39	32.3 ± 2.6
16	200	20	1:12	4.73 ± 0.75	22.65 ± 1.12	24.6 ± 3.4
17	400	20	1:12	4.76 ± 0.49	15.94 ± 0.78	19.3 ± 3.7

**Table 2 foods-10-00667-t002:** The physicochemical properties of yogurt without soy protein isolate (SPI) (control), or with SPI and high pressure treated SPI (HHP-SPI) (8%).

		Control	SPI Yogurt	HHP-SPI Yogurt
Final pH		4.15 ± 0.02 ^a^	4.56 ± 0.05 ^b^	4.53 ± 0.03 ^b^
titration acidity (°T)		67.2 ± 0.4 ^b^	63.6 ± 0.6 ^a^	64.1 ± 0.3 ^a^
WHC (%)		56.37 ± 1.71 ^a^	83.42 ± 0.96 ^b^	90.22 ± 2.83 ^c^
color	L* ^1^	80.70 ± 0.02 ^c^	74.02 ± 0.01 ^a^	74.64 ± 0 ^b^
	A* ^2^	−1.89 ± 0.01 ^a^	0.33 ± 0 ^c^	−0.18 ± 0.01 ^b^
	B* ^3^	10.30 ± 0.02 ^a^	15.21 ± 0.01 ^c^	13.66 ± 0.01 ^b^
	ΔE	-	8.58 ± 0.01 ^b^	7.13 ± 0.01 ^a^

^a–c^ Different superscript letters within the same column indicated significant difference (*p* < 0.05). ^1^ lightness value; ^2^ Red-green axis value; ^3^ Yellow-blue axis value.

**Table 3 foods-10-00667-t003:** The model fitting results of yogurt without SPI (control) or with SPI and HHP-SPI.

Samples	Shear Rate Rise			Shear Rate Drop		
	n	k(Pa*s^n^)	*R* ^2^	n	k(Pa*s^n^)	*R* ^2^
control	0.39 ± 0.01 ^c^	32.35 ± 0.13 ^a^	0.98	2.35 ± 0.01 ^c^	15.83 ± 0.02 ^c^	0.98
SPI	0.23 ± 0.01 ^a^	41.87 ± 3.08 ^b^	0.99	0.80 ± 0.01 ^b^	3.84 ± 1.41^a^	0.99
HHP-SPI	0.31 ± 0.01 ^b^	58.85 ± 6.11 ^c^	0.98	0.77 ± 0.01 ^a^	9.53 ± 4.88 ^h^	0.98

^a–c^ Different superscript letters within the same column indicated significant difference (*p* < 0.05).

## Data Availability

No new data were created or analyzed in this study. Data sharing is not applicable to this article.

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
