# Peer review of "Optimization of High Hydrostatic Pressure Treatments on Soybean Protein Isolate to Improve Its Functionality and Evaluation of Its Application in Yogurt"

_foods, 2021, doi:10.3390/foods10030667_

Round 1

Reviewer 1 Report

The work foods-1124513 aimed to improve the functional properties of soybean protein isolate (SPI) by high hydrostatic pressure (HHP) optimized by Response surface methodology (RSM) with the last aim to develop a SPI-milk mixed yogurt. The manuscript has been well written but needs some improvements. One thing I particularly dislike about works using RSM is that Authors only focus on the statistical results (the different statistical models produced by the method) and leave behind the phenomenology involved in the studied material. For this reason, my main recommendation to the authors of the current work is that they relate these statistical models to the phenomenology involved, since these empirical equations usually only serve to describe specific situations, such as those of this particular study. Below I will leave in detail the changes that I consider should be revised:

Major considerations:

- The use of HHP treatment to improve the functional properties of SPI is not new [1–5]; however, I find the novelty of this work in the use of SPI to produce a soy-based yogurt. I believe that the authors should specify this.

- The manuscript is supported by a large number of analyses, which, however, should be discussed more from a phenomenological and less statistical point of view.

- I strongly recommend that the quality of the figures be improved.

- Figure 4: The inner table, which I consider should not be part of the Figure, is not clearly visible. In reference to this Figure, I recommend the authors to discuss more of the relationship between the Flow curves and the structure of the beverage. It would have been good for it a support with SEM; but the authors can discuss a little more of it instead of going deeper into the models.

- Solubility and WHC should be discussed in greater depth and beyond the statistical models, as these give guidance to the phenomenology to be explained in the manuscript. I consider that these two properties should be discussed as a function of pH (which apparently was not studied by the authors), since evidently the protein stability of SPI will be at the mercy of this physicochemical variable.

- L63: Report the limited data.

- L77 (2.2 HHP treatments): The authors should further detail the methodological section of HHP treatments or cite studies where the same equipment has been used and more details are given.

- L97 and 105 (2.4.1 Solubility and 2.4.2 Water holding capacity (WHC) and oil holding capacity): Describe the pH of every medium used.

- L306-309: What is the basis for this assertion?

- Conclusions: What are the legal conditions of applicability of the milk beverage?

Minor considerations:

- - L100: 1 h. Regarding this and given the nature of the journal I recommend implementing the guideline from the National Institute of Standards and Technology (NIST). For this purpose, it should be noted that each numerical value must be accompanied by its respective unit and, in addition, this unit must be separated by a space from the numerical value (i.e. 10 °C, 20 °C and 30 °C).

- A section on abbreviations should be included.

- When reporting p-values in journals, it is common practice for p to be lowercase and italicized, e.g., p≤0.05.

References

  1. Puppo, C.; Chapleau, N.; Speroni, F.; de Lamballerie-Anton, M.; Michel, F.; Añón, C.; Anton, M. Physicochemical Modifications of High-Pressure-Treated Soybean Protein Isolates. J. Agric. Food Chem. 2004, 52, 1564–1571, doi:10.1021/jf034813t.
  2. Xi, J.; He, M. High hydrostatic pressure (HHP) effects on antigenicity and structural properties of soybean β-conglycinin. J. Food Sci. Technol. 2018, 55, 630–637, doi:10.1007/s13197-017-2972-2.
  3. Li, H.; Zhu, K.; Zhou, H.; Peng, W. Effects of High Hydrostatic Pressure on Some Functional and Nutritional Properties of Soy Protein Isolate for Infant Formula. J. Agric. Food Chem. 2011, 59, 12028–12036, doi:10.1021/jf203390e.
  4. Liu, D.; Zhang, L.; Wang, Y.; Li, Z.; Wang, Z.; Han, J. Effect of high hydrostatic pressure on solubility and conformation changes of soybean protein isolate glycated with flaxseed gum. Food Chem. 2020, 333, 127530, doi:10.1016/j.foodchem.2020.127530.
  5. Manassero, C.A.; Vaudagna, S.R.; Añón, M.C.; Speroni, F. High hydrostatic pressure improves protein solubility and dispersion stability of mineral-added soybean protein isolate. Food Hydrocoll. 2015, 43, 629–635, doi:10.1016/j.foodhyd.2014.07.020.

Reviewer 2 Report

Manuscript Number: foods-1124513

Title: Optimization of high hydrostatic pressure treatments on soybean protein isolate to improve its functionality and evaluation of its application in yogurt

Journal: Foods

General comments

The study reports on the optimization of high hydrostatic pressure processing to modify the functionality of soybean protein isolate. The authors did a very good job at presenting and analyzing the results. The text was easy to follow. The study may help with the current efforts to improve functionality of plant-based proteins. There are some aspects of the study that were identified and required immediate attention. One of the concerns is that the RSM was optimized for many variables (some of them are not even relevant such as the harness of gel). Moreover, the initial properties of the SPI used in the study were not described (water holding capacity, emulsifying activity index, solubility, etc.) and how these properties compared to the ones of the optimized HHP-SPI.  Out of the 52 references cited, ten (10) are more than 15 years old; so the reviewer encourages the authors to use recent references throughout the text (preferably references that are less than 10 years old).

Introduction

Line 29 – Please include an statement on how SPI are produced?

Line 31 – What type of solvents?

Please include similar studies that have modified plant-based proteins using HHP

Please include an introduction statement for the optimization method and explain why RSM was used over other methods?

Materials and Methods

Line 81 – 85 – Detailed description of sample preparation is needed, how SPI was mixed with water, how much sample was treated at a given time? Where the freeze dried samples grounded?, if so, described the procedure.

Line 87 -90. What are the units of the response variables?

Line 90 – Details on how the treatments were selected using a box-benhnken design, what are the details of the software (developer, country, etc.)

Line  142- 145 Add more details of particle size and zeta potential determination.

Line 156 – 158 – What are the units for protein hydrophobicity?

Line 165 – 167 – What are the units of surface sulfhydryl group content?

Line 184 – 185 – What type of HHP treated SPI was used for yogurt making?

Results and Discussion

The regression models presented are not optimized, if the regression coefficients have a P-value >0.05; they should be removed from the regression model. Regression models should be revised.

Harness of gel is even a variable for consideration? It should be removed from the analysis. It seems that the results for hardness of gel are the same for all of the evaluated conditions (Table 1).

What are the initial properties of SPI?

Tables and Figures

In general letters and numbers in figures are too small and difficult to read.

Figure 2 – Add letters to show statistical significance

Figure 3 – Add letters to show statistical significance

Figure 4 – The words and numbers presented in table embedded in the figure is very difficult to read. Please revise.

Figure 5 – Please include SD bars and letters to show statistical significance

Author Response

Response to Reviewer 2 Comments

Point 1: The study reports on the optimization of high hydrostatic pressure processing to modify the functionality of soybean protein isolate. The authors did a very good job at presenting and analyzing the results. The text was easy to follow. The study may help with the current efforts to improve functionality of plant-based proteins. There are some aspects of the study that were identified and required immediate attention. One of the concerns is that the RSM was optimized for many variables (some of them are not even relevant such as the harness of gel). Moreover, the initial properties of the SPI used in the study were not described (water holding capacity, emulsifying activity index, solubility, etc.) and how these properties compared to the ones of the optimized HHP-SPI.  Out of the 52 references cited, ten (10) are more than 15 years old; so the reviewer encourages the authors to use recent references throughout the text (preferably references that are less than 10 years old).

 Response 1: Thanks very much for taking your time to review this manuscript. I really appreciate all your comments and suggestions! Please find my itemized responses in below and my revisions/corrections in the re-submitted files. The initial properties of the SPI used in the study were described in 3.2 as 0.1 MPa group. Three 15-year-old references has been removed.

Point 2: Line 29 – Please include an statement on how SPI are produced?

Response 2: SPI is generally produced through the method by collecting as precipitated curd in the acidic condition, e.g. pH 4.5. It has been added in the manuscript (Line 30).

Point 3: Line 31 – What type of solvents?

Response 3: This sentence was removed from the manuscript due to ambiguity.

Point 4: Please include similar studies that have modified plant-based proteins using HHP

Response 4: 5 more studies related to HHP modifying soy proteins have been cited and discussed (Line 41-51).

Point 5: Please include an introduction statement for the optimization method and explain why RSM was used over other methods?

 Response 5: A major drawback with other approaches is time-consuming and does not take the interdependence of experimental factors into account. The response surface methodology (RSM) is a practical technique for testing multiple process variables, and fewer experimental trials are needed comparing with single factor experiment. RSM is useful for optimizing, designing, developing, and improving processes where the responses are affected by several variables. Moreover, interactions between variables can be identified and quantified by RSM and it is widely used in food industry processes for optimizations. (Line 91-94)

Point 6: Line 81 – 85 – Detailed description of sample preparation is needed, how SPI was mixed with water, how much sample was treated at a given time? Where the freeze dried samples grounded?, if so, described the procedure.

Response 6: The procedure about the sample preparation has been revised and more details have been added. (Line 83-88)

Point 7: Line 87 -90. What are the units of the response variables?

Response 7: The units of the response variables has been added. (Line 95-96)

Point 8: Line 90 – Details on how the treatments were selected using a box-benhnken design, what are the details of the software (developer, country, etc.)

Response 8: Treatment were selected by literature review and pre-test. Details of the software has been added. (Line 96)

Point 9: Line 142- 145 Add more details of particle size and zeta potential determination.

Response 9: This part was removed from the manuscript because these two properties are not closely related to the improvement of yogurt properties.

Point 10: Line 156 – 158 – What are the units for protein hydrophobicity?

Response 10: Generally, articles using this method have not report the unit. It is calculated by linear regression analysis as the initial slope of relative fluorescence intensity versus protein concentration. There are some references using same method:

  1. Xi, J.; He, M. High hydrostatic pressure (HHP) effects on antigenicity and structural properties of soybean β-conglycinin. J. Food Sci. Technol. 201855, 630–637, doi:10.1007/s13197-017-2972-2.
  2. Li, H.; Zhu, K.; Zhou, H.; Peng, W. Effects of High Hydrostatic Pressure on Some Functional and Nutritional Properties of Soy Protein Isolate for Infant Formula. J. Agric. Food Chem. 201159, 12028–12036, doi:10.1021/jf203390e.
  3. Wang, X.; Tang, G.; Li, B.; Yang, X.; Li, L. and Ma, C. Effects of high-pressure treatment on some physicochemical and functional properties of soy protein isolates. Food Hydrocoll., 2008, 22(4), 560–567. doi: 10.1016/j.foodhyd.2007.01.027

Point 11: Line 165 – 167 – What are the units of surface sulfhydryl group content?

Response 11: The method has been rewritten and the unit has been added. (Line 148-158)

Point 12: Line 184 – 185 – What type of HHP treated SPI was used for yogurt making?

Response 12: We used the the SPI treated under the optimized HHP parameters. (Line 176)

Point 13: The regression models presented are not optimized, if the regression coefficients have a P-value >0.05; they should be removed from the regression model. Regression models should be revised.

Response 13: The regression models has been revised. (Eqs. 9-11 and Table S1)

Point 14: Harness of gel is even a variable for consideration? It should be removed from the analysis. It seems that the results for hardness of gel are the same for all of the evaluated conditions (Table 1).

Response 14: Gel properties of SPI has been removed from the manuscript because heat-induced gel properties of SPI are not closely related to the improvement of yogurt properties.

Point 15: What are the initial properties of SPI?

Response 15: The initial properties of SPI has been mentioned in the discussion as 0.1MPa (Atmospheric pressure) treatment group. (Figure 2 and 3)

Point 16: In general letters and numbers in figures are too small and difficult to read.

Response 16: The quality of figures has been improved. (Figure 2-5)

Point 17: Figure 2 – Add letters to show statistical significance

Response 17: Letters has been added in Figure 2.

Point 18: Figure 3 – Add letters to show statistical significance

Response 18: Letters has been added in Figure 3.

Point 19: Figure 4 – The words and numbers presented in table embedded in the figure is very difficult to read. Please revise.

Response 19: The table and the figure has been separated. Please see Table 4 and Figure 4.  

Point 20: Figure 5 – Please include SD bars and letters to show statistical significance

Response 20: Letters and SD bars has been added in Figure 5.

Round 2

Reviewer 1 Report

The first and second reply do not specify in which part of the corrected text the changes were made.

The lines to which the authors refer in their response letter do not correspond to the corrections made to the resubmitted manuscript, so their corrections cannot be appreciated.

Finally, I consider that the quality of the Figures has not been improved.
